# Health Effects of Dust Storms on the South Edge of the Taklimakan Desert, China: A Survey-Based Approach

**DOI:** 10.3390/ijerph19074022

**Published:** 2022-03-28

**Authors:** Aishajiang Aili, Hailiang Xu, Xinfeng Zhao

**Affiliations:** State Key Laboratory of Desert and Oasis Ecology, Xinjiang Institute of Ecology and Geography, Chinese Academy of Sciences, Urumqi 830011, China; aishajiang@ms.xjb.ac.cn (A.A.); zxinfeng668@sohu.com (X.Z.)

**Keywords:** dust weather, air pollutants, symptoms, health effects, PCA, Taklimakan desert

## Abstract

Dust storms have already become the most serious environmental problem on the south edge of the Taklimakan desert because of their frequent occurrences. To investigate the health effects of dust storms on public health in Moyu County, one of the most severe dust-storm-affected areas located at the south edge of the Taklimakan desert, China, primary data were collected from 1200 respondents by using a questionnaire survey for 15 health symptoms. The data were analyzed by comparing the mean tool (independent t-test and ANOVA) and the severity of different symptoms among different age groups. Principal component analysis (PCA) was applied to further analyze the multivariate relationships between meteorological factors, dust storm intensity, air pollution level, and severity degree of the different symptoms. The results show that significant correlations exist between dust storm intensity, air pollutants (PM_2.5_, PM_10_, O_3_, SO_2_, NO_2_, and CO), meteorological factors, and health symptoms. During dusty weather, no matter the age group, the number of respondents who suffered from different health symptoms was higher compared to non-dusty days. Three types of dusty days were considered in this study: suspended dust, blowing dust, and sand storms. The impacts of sand storm weather on public health are stronger than those from blowing dust weather, suspended dust weather (haze), and non-dust weather. The people in the age groups above 60 years and below 15 years were more sensitive to different dust weather than people in the age groups between 15 and 60. “Dry throat with bitter taste”, “Depression”, “Dry and itchy throat”, and “Mouth ulcer” are the main symptoms caused by dust storms.

## 1. Introduction

Taklimakan desert, the world’s second largest shifting sand desert with a total area of 337,000 km^2^, is one of the major source regions of dust storms in China [1], and its surrounding area has already become the most severe dust-storm-affected area due to the high frequency of dust storms [2,3,4]. During the dust storm period, large amounts of sand and dust particles greatly deteriorate air quality and influence people’s health, and some viruses and toxic pollutants, such as the heavy metals, bacteria, and poisonous minerals blown by the wind and transported over long distances, are inhaled by people into their respiratory tracts and lungs and cause severe health effects [5,6,7]. Small particles with a diameter of 0.5–5 microns can be deposited directly in human lungs through the respiratory tract, causing respiratory diseases, and can also be absorbed into blood circulation, causing other organ diseases [8,9,10].

The impacts of dust storms on human health are closely related to the fine particles they carry [11]. There are many reports about the impact of particulate air pollutants on human health. Zhang et al. studied the health effects of air pollutants on the residents of Lanzhou in northwest China and found that there was a lag effect of air pollutants on daily cardiovascular disease admissions and daily deaths. The lag effect on total cardiovascular disease, ischemic cardiovascular disease, hypertension, and death was different with various pollutants. Jimenez et al. analyzed the short-term impact of particulate matter (PM_2.5_) on daily mortality among the over 75 age group in Madrid (Spain) and found that PM_2.5_ and PM_10_ have a significant short-term impact on mortality. Xu et al. studied the impact of air temperature on children’s health and indicated that hot and cold temperatures mainly affect cases of infectious diseases among children, including gastrointestinal diseases, malaria, respiratory diseases, and hand, foot, and mouth disease. Pediatric allergic diseases, such as eczema, are also sensitive to temperature extremes [12,13,14,15]. Studies have found that particulate air pollutants can cause acute and chronic bronchitis, asthma, pneumonia, and even lung cancer and other respiratory and cardiovascular diseases [12]. Due to direct contact with the desert, it is easy to form dusty weather even by weaker wind conditions in Moyu County. Local people in the study area were vulnerable to dust storm impacts and had already realized its adverse effects on the environment, transportation, and public health. However, there is a lack of scientific investigation on the association between dust storms and human health in severe dust storm areas. Only some preliminary epidemiologic studies have been conducted on the fine sand and dust particles causing respiratory diseases [13,14,15]. How do dust storms affect human health? Which age groups are more vulnerable to dust storms? What is the relationship between dust storm intensity, air pollution level, and health symptoms? These problems need to be further studied.

It is therefore important to carry out related research work on the effects of dust storms on public health in severe dust-storm-affected areas. This study, therefore, was designed to partly fill in this information gap. The primary data on health symptoms, which were collected by door–to–door surveys during March–June 2018 in Moyu County, were analyzed by comparing mean analysis and PCA [16,17,18]. The exposure–response relationship between health symptoms, dust storm intensity, meteorological factors, and air pollutants concentration were examined, while the severity of different symptoms among different age groups was investigated. To our knowledge, this study is the first health impact research that has ever been carried out in this area. Research focusing on this dust-storm-vulnerable region will help to understand the relationship between dust storms and health symptoms, which can provide the basis for mitigating the negative effects of dust storms on human health.

## 2. Materials and Methods

### 2.1. Description of the Study Area

Moyu County is located at the southwest of the Xinjiang Uygur Autonomous Region, the north slope of Kunlun Mountain, and the south edge of the Taklimakan desert. Its geographical coordinates are 36°36′~39°38′ N, 79°08′~80°51′ E. The total area of Moyu County is 25,788.86 km^2^, and it includes 16 town/townships, with a total population of 632,740 [19]. The topography of Moyu County is high in the south and low in the north, with an altitude of between 1120 and 3663 m. Its southern part is a mountainous area with undulating hillsides and its central part is a flood alluvial fan plain. Moyu County belongs to a warm, temperate, dry, desert climate, with distinct seasons: summer is hot and dry, with less rain, and temperatures rise rapidly in the spring. The main meteorological characteristics of Moyu County include sparse precipitation, abundant light, a long and frost-free period, and a large temperature difference between day and night. The annual average temperature is 11.3 °C, the monthly average temperature shows its highest value in July (26.4 °C) and lowest in January (−6.5 °C), the extreme minimum temperature is −18.7 °C, the annual average precipitation is 36–37 mm, the annual evaporation is 2239 mm, the frost-free period is 177 days, and the annual sunshine time is 2655 h [20].

The dry climate and large area of desert provide favorable conditions for frequent dust weather. The annual dusty days in Moyu County are more than 220, and the strong sand storm weather can reach about 60 days. In this study, dust storm weather was classified into three levels, depending on the severity, using the criteria given by AQSIQ/NSC (2006) [21]. The weaker type of dust storm is called suspended dust weather and refers to the suspended dust in the air under calm or low wind conditions. The medium to severe type of dust weather is called blowing dust, with horizontal visibility ranging from 1 km to 10 km. The most strong dust weather is called a sand storm (sometimes it is simply called a dust storm), which refers to such phenomena when the instantaneous wind velocity is over 25 m/s and horizontal visibility is below 1 km [3,21]. The number of dusty days in spring and summer account for approximately 90% of the total number of dust storm days in a year. The high-occurrence period of dust storms is from April to August, and May and June are the most active periods of dust storms.

### 2.2. Material Sources

Air pollution and meteorological data: In this study, local meteorological data including daily average temperature, wind speed, and types of dusty weather in the period of 1 March to 30 June 2018 were obtained from the Moyu Meteorological Station, as well as the China Meteorological Data Sharing Service System (http://cdc.cma.gov.cn, 1 August 2019). Air pollution indexes (APIs) as well as air quality data, including daily average concentrations of PM_2.5_, PM_10_, SO_2_, NO_2_, O_3_, and CO, were obtained from the Environmental Monitoring Center of the Xinjiang Uyghur Autonomous Region (XUAR) and the Hotan Environmental Protection Bureau, respectively. At present, China mainly adopts two methods to collect air pollutant concentrations: sampling and weighing based on filter membrane and continuous automatic measurement by using a PM automatic component analysis device (PX-375) and a dynamic calibrator (APMC-370).

Survey data: To examine the effects of different types of dust weather on public health, primary data were collected from the study area using a questionnaire form. The investigation of this study was carried out after receiving the approval of Jahanbagh township’s government and local school leaders. The data collection period was 122 days, from 1 March to 30 June 2018, which covered the dusty season. A total of 1200 respondents were selected from the Jahanbagh town of Moyu County (Figure 1). The number of respondents was determined by using multistage statistical sampling design [22], and respondents were selected from the different age groups. The steps used to calculate sample size are as follows.

#### 2.2.1. Step 1: Base Sample-Size Calculation

For a survey design, based on a simple random sampling, the sample size required can be calculated using following equation:n = [t^2^ × p (1 − p)]/m^2^
where n = the required sample size, t = confidence level at 95% (standard value of 2.4), p = the estimated prevalence of dust storms affecting people in the study area, and m = the margin of error at 5% (0.05).

To give the largest sample size (n), p was assigned to be 0.5, i.e., it was roughly estimated that 50% of the local people have health symptoms because of dust storms (Qian et al., 2005).

Thus, n = [2.4² × 0.5 (1 − 0.5)]/0.05² = 576.

#### 2.2.2. Step 2: Design Effect

The anthropometric survey is designed as a cluster sampling (a representative selection of villages/communes), not a simple random sampling. To correct for the difference in design, the sample size is multiplied by the design effect. The design effect is generally assumed to be two for surveys using the cluster-sampling method (Magnani, 1997). Thus, the value of n would be 1152.

#### 2.2.3. Step 3: Contingency

The sample size is further increased by 5% to account for contingencies such as non-response or recording error; hence, n would be 1199.7, or roughly = 1200.

#### 2.2.4. Step 4: Distribution of Observations

Based on the shares of population within a particular age range (group) in the total population, the number of the respondents for each age group is proportionally distributed by using the following formula: ni = (Pi/P)/1200
where, n_i_ is the number of respondents in different age groups, P_i_ is the total population of each group, and P is the total population in the study area. For example, the total number of children in this area is 11,092, which accounts for around 30.04% of total population (36,470).

Therefore, the number of respondents for children is n_1_ =1200 × (11,092/36,470) = 365. A total of 365 respondents were selected from a local school and equally distributed for each grade, from grade 1 (6 years old) to grade 9 (14 years old). In the same way, the number of respondents for other age groups can also be calculated, e.g., n_2_ = 185, n_3_ = 455, or n_4_ = 195.

Finally, for the age-based distribution, 365 respondents were selected from children (under the age of 15), 185 respondents were selected from young people (15–25), 455 respondents were selected from adult (25–60), and 195 respondents were selected from elder people (above age of 60). The survey forms were distributed to the respondents at the end of February. To ensure the completeness and accuracy of the questionnaire data, during the data collection period, 20 volunteers from study area were selected for the distribution and collection of the questionnaire form. In addition, for the children (under the age of 15), parents and teachers were requested to help them to fill the forms. Fifteen types of symptoms, including dry throat with bitter taste (S1), tears (S2), runny nose (S3), sneeze (S4), dry eyes (S5), shortness of breath (S6), chest tightness (S7), cough (S8), depression (S9), expectoration (S10), stuffy nose (S11), dry and itchy throat (S12), hoarseness (S13), cleft lip (S14), and mouth ulcer (S15) were listed in the form. Respondents were required to fill the form with the symptoms that they experienced each day, with four levels of severity (negligible, somewhat, medium, and serious).

### 2.3. Statistical Analysis

The collected data of daily average air pollutant concentration, daily average temperature, wind speed, and the occurrence frequencies of different dust weather types during the study period were analyzed by the SPSS (Statistical Package of Social Science) software to reveal the relationships between air pollution levels and different dust weather conditions (normal days, suspended dust weather, blowing dust weather, and sand storm weather).

The questionnaire survey data collected from the 1200 respondents during the study period (total of 122 days) were entered into SPSS software, and the daily average severity of each symptom was calculated based on the different severity degree, including negligible (encoded 0), somewhat (encoded 1), medium (encoded 2), and serious (encoded 3), by using following equation:(1)$$S_{k}=\frac{\sum_{i=1}^{n}x_{i}}{3n}\times 100\%$$

In this equation, *S_k_* is the average severity of symptoms in a selected day, n is the number of respondents, and *x_i_* is the severity degree of symptoms in a selected day (*x_i_* = 0, 1, 2, or 3). The data then were analyzed separately for the four age groups, as well as for four different weather conditions.

To reveal multivariate relationships between dust storm intensity, air pollutants, meteorological factors, and severity of the different symptoms, a total of 26 variables including daily average severity of the 15 types of considered symptoms, daily dust storm intensity (the quantification value of a non-dusty day was 0, suspended dust was 1, blowing dust was 2, and sand storm was 3), daily average air pollutant concentration (six pollutants), and daily average wind speed, air humidity, temperature, and air pressure (four variables) were analyzed by using PCA. The variables first were converted to standard scores, which have a mean of 0 and a standard deviation of 1, by using following equation:

Standard value = (original data − mean)/standard deviation.

In this study, PCA with a Varimax rotation was applied and the PC score was obtained by using following equation. For example, let X = [xi] be any k × 1 random vector. We now define a k × 1 vector Y = [*y_i_*], where for each i the i^th^ principal component of X is
(2)$$y_{j}=\overset{k}{\underset{j=1}{\sum }}\beta_{ij}x_{j}$$
where, *β_ij_* is a regression coefficient, and since each *y_i_* is a linear combination of the *x_j_*, Y is a random vector.

## 3. Results and Discussion

### 3.1. Daily Average Concentration of Air Pollutants in Different Dust Weather

During the study period, total of 122 days from 1 March to 30 June 2018, sand storm weather occurred for 8 days, blowing dust weather occurred for 33 days, suspended dust weather occurred for 46 days, and the other 34 days were normal/non-dust days. As expected, for both dust storm frequency and intensity, May showed a peak, with a total frequency of 27 times (9 days of suspended dust, 14 days of blowing dust, and 2 days of sand storms), and June showed the lowest number, with a frequency of 16 times. The daily average concentrations of PM10 and PM2.5 in dusty days were higher than in non-dusty days (normal/clean days). The pollutant levels increased with the severity of dust storms. The daily average concentrations of PM10 and PM2.5 showed a consistent trend and are ranked in the following order: sand storm > blowing dust > suspended dust > normal days, which indicated that dusty weather can significantly impact air pollutant concentrations. For example, during the normal days, the average levels of PM_10_ and PM_2.5_ were, respectively, 196.42 ± 21.4 μg/m^3^ and 162.41 ± 14.2 μg/m^3^, as compared to the corresponding levels observed during the suspended dust weather days of 260.11 ± 26.4 μg/m^3^ and 211.09 ± 16.2 μg/m^3^, the blowing dust weather days of 297.42 ± 21.4 μg/m^3^ and 239.41 ± 14.2 μg/m^3^, and the sand storm weather days of 405.12 ± 39.2 μg/m^3^ and 325.48 ± 29.3 μg/m^3^ (Figure 2).

Overall, the variations of PM10 and PM2.5 had similar trends, and they all peaked during the sand storm weather days. Unlike PM10 and PM2.5, the high levels of SO_2_, NO_2_, O_3_, and CO during the dust storm weather were not initially expected.

### 3.2. Health Symptoms in Different Dust Weather Conditions

The survey data were analyzed for four different dust weather conditions. The average severity degree of the 15 health symptoms during the dusty days and non-dusty days were analyzed to investigate the effects of dust weather on public health, and the results are presented in Figure 3.

It can be clearly seen from Figure 3 that there is a significant difference in the severity of all kinds of symptoms between non-dust weather and dust weather (suspended dust, blowing dust, and sand storms). The severity of all types of health symptoms shows its lowest value in the non-dusty days and increased values with dust storm intensity. The average severity of all types of symptoms is 22.7% for non-dusty days, 24.2% for suspended dust weather, 30.6% for blowing dust weather, and 33.8% for sand storm weather. During the dusty days, no matter the age group, the number of respondents who suffered from different health symptoms was higher as compared to the non-dusty days. In other words, the more severe the dust weather, the higher the severity of symptoms. However, the severity degree of some symptoms, such as “Shortness of breath”, “Chest tightness”, “Cleft lip”, and “Mouth ulcer” in “non dusty days” is similar to “suspended dust weather”, while “blowing dust weather” is similar to “sand storm days”. During the suspended dust weather, the air visibility, dust particle concentrations, and wind speed are not much different than the non-dusty days. Therefore, the severity degree of such kinds of severe symptoms shows a similar value in two kinds of weather conditions. However, in blowing dust and sand storm weather, the air visibility, dust particle concentrations, and wind speed increase significantly. Although there is a certain gap between two types of dust weather, the severity degree of these symptoms shows a similar value. Our survey data is based on the respondents’ own feelings about various symptoms. Therefore, some data are not necessarily the same as the general trend. Comparing the severity degree of different symptoms, dry throat with bitter taste (31.9% for suspended dust weather, 41.2% for blowing dust weather, and 40.9% for sand storm weather), depression (38.3% for suspended dust weather, 40.2% for blowing dust weather, and 47.7% for sand storm weather), dry and itchy throat (32.5% for suspended dust weather, 42.7% for blowing dust weather, and 43.9% for sand storm weather), and mouth ulcer (26.9% for suspended dust weather, 40.8% for blowing dust weather, and 41.1% for sand storm weather) are more serious symptoms with a higher severity degree than others. Further, the severity of different symptoms was analyzed separately for different age groups (Figure 4).

It can be seen from Figure 4 that dry throat with bitter taste, depression, dry and itchy throat, and mouth ulcer were the four serious symptoms observed with a higher severity degree (≥30%) for all the age groups during the study period. Compared with the other age groups, elder people (≥60 years) were more sensitive to dusty weather. The severity of all types of symptoms for elder people shows the highest value, with the average severity of 34.6%. Children (under 15 years) were also more sensitive to dusty weather than young people and adults. The severity of the symptoms of dry throat with bitter taste (34.2%), tears (26.6%), runny nose (33.3%), cough (37.5%), expectoration (31.1%), and stuffy nose (19.9%) shows a higher value than young people and adults, respectively. The severity of the symptoms of sneeze and stuffy nose shows a lower value for all age groups (≤30%). Compared with other age groups, adults (25–60) were less sensitive to dust weather. In other words, the severity degree of all symptoms shows the lowest score compared to that of the other age groups. This is because adults are in good physical condition, so the incidence of those symptoms is a little less than that of people of other ages.

Overall, the dust weather has had a significant impact on public health in the study area and across all age groups. Elder people (above 60 years) and children (below 15 years) were more sensitive to dust weather than the other age groups. This may be explained by the differences in other conditions present in different age groups, such as physical condition, personal activities, and so on.

### 3.3. Multivariate Relationship between Air Pollutants, Meteorological Factors, and Health Symptoms

PCA was applied to reveal the relationships between the different health symptoms, dust storm intensity, air pollutant levels, and meteorological variables during the 122 days of the survey period. The dataset for PCA consisted of 26 variables including daily dust storm intensity, daily average severity of the 15 health symptoms, air pollutant concentrations (PM_2.5_, PM_10_, O_3_, SO_2_, NO_2_, and CO), and four meteorological factors (daily average temperature, wind speed, air humidity, and air pressure). The variables first were converted to standard scores, which have a mean of 0 and a standard deviation of 1. The results of the PCA are shown in Table 1.

Four PCs were extracted, which collectively explained 71.3% of the total variance in the original data set. The loadings of different variables on each PC are presented in Table 1. PC1 (explaining 37.6% of the total variance) shows high loadings (above 0.40) of dust storm intensity, temperature, wind speed, PM_10_, and PM_2.5_ and the health symptoms of “dry throat with bitter taste”, “dry eyes”, “depression”, “itchy throat”, and “hoarseness”. High loadings of dust storm intensity, meteorological factors, air pollutants, and health symptoms on a particular PC indicate that the symptoms were likely related to dust storms and these pollutants. As discussed in Section 3.1 of this paper, both dust storm frequency and intensity show the highest values in May, when temperature and wind speed were also increased, and the PM_10_ and PM_2.5_ concentrations were also at their highest values in this month. Particulate matter has a small size and a relatively large surface area, which makes it easy to absorb microorganisms in the air. Wind speed, temperature, and other meteorological factors are also known to be associated with the mortality and morbidity associated with respiratory diseases. “Dry throat with bitter taste”, “itchy throat”, and “hoarseness” belong to upper respiratory tract diseases, as eyes and throats are always very directly affected by dust weather. “Depression” is an emotional symptom, but is mostly caused by the low visibility of air during dust weather.

PC2 (explaining 14.5% of the total variance) confirms the association between dust storm intensity (negative value), air humidity, air pressure, SO_2_, NO_2_, CO, and the health symptoms of “tears”, “runny nose”, “cough”, and “expectoration”. At the end of winter and in early spring, e.g., the month of March, in study area, both dust storm frequency and intensity were lower than in other months, but the air pressure and air humidity were higher due to the low temperature in this period. Coal burning in this heating period made significant contributions to the concentrations of SO_2_, NO_2_, and CO. The health symptoms of “tears”, “runny nose”, “cough”, and “expectoration” may not be caused by dust weather but were likely related to the low temperature and higher concentration of toxic air pollutants (SO_2_, NO_2_, and CO). Our findings are consistent with the results of other similar studies. Tian et al. analyzed the spatial patterns of hospitalization for chronic lung diseases and explored associations with PM_10_, SO_2_, NO_2_, mean temperature, and relative humidity at the district level in Beijing by using Poisson regression models, and found that the effect of mean temperature, SO_2_, and NO_2_ were significant, with a higher relative risk of 1.071, 1.092, and 1.107 [6]. Further research needs to be conducted with more detailed information on environmental conditions, personal exposure, genetic predispositions, and so on.

PC3 (explaining 10.4% of the total variance) shows the association between wind speed, PM_10_, and PM_2.5_ and the health symptoms of “shortness of breath”, “chest tightness”, “stuffy nose”, “cleft lip”, and “mouth ulcer”. The wind speed actually is related directly to the severity of dust weather, and covariates with PM_10_ and PM_2.5_ concentrations, as discussed above. The health symptoms of “shortness of breath” and “chest tightness” belong to serious respiratory diseases. These diseases are mostly caused by strong wind during dust weather. During sand storm weather, strong wind can also directly attack people’s eyes, noses, and mouths, causing the health symptoms of “stuffy nose”, “cleft lip”, and “mouth ulcer”.

PC4 (explaining 8.8% of the total variance) shows high loadings of temperature and O_3_ and the health symptoms of “sneeze” and “dry eyes”. Both dust storm frequency and intensity increased in late spring and early summer, e.g., the month of May, when the temperature also increased. Based on the previous study [23], concentrations of O_3_ will increase with the temperature. As a toxic air pollutant, a higher concentration of O_3_ most likely caused the health symptoms of “sneeze” and “dry eyes”.

To our knowledge, this study is the first research analyzing the health impacts of dust storms in this area. The findings of this study are consistent with the results of other similar studies which have been carried out in other places in the world. Many epidemiological studies have shown that short-term increases in ambient air pollutants and dust weather frequencies are associated with an acute rise in hospital visits. Zhang et al. found a significant association between dust weather and the number of hospital admissions for patients with respiratory diseases during spring in Lanzhou, China [15]. Similarly, research conducted by Zhang et al. in Beijing showed the association between air pollution and respiratory system diseases using time–series analysis [7]. Meng et al. analyzed the influence of dust storms on daily respiratory and circulatory outpatient numbers in Xi’an, northwest China, by using the generalized additive model (GAM), and found that the number of respiratory and circulatory outpatients increased with dust storm intensity [10].

Overall, dust weather was shown to induce significant impacts on public health in the study area, across all age groups. However, there is no special institutional organization for natural disaster management in the study area, and only the local meteorological station and Environmental Protection Bureau assumed some responsibilities for forecasting service and some technical supports. Moreover, current dust storm warnings in this area appear to be provided via local television and internet and are accessible from mobile phones. Predicting dust storms in time obviously helps people to be better prepared. The results of this study can be used to raise awareness so that actions can be taken to avoid the excessive exposure of people to dust storms.

## 4. Conclusions

To our knowledge, this study is the first health impact research that has ever been carried out in this area. Regarding the results of our study, our findings were consistent with the results of other similar studies which have been carried out in other places in the world. Many epidemiological studies have shown that short-term increases in ambient air pollutants and dust weather frequencies are associated with an acute rise in hospital visits. As a matter of fact, other environmental factors and individuals’ physical conditions also play important roles, and the health symptoms may not be entirely caused by dust weather and air pollutants. The main conclusions of our research are as follows:Both dust storm frequency and intensity reached their highest values in May and the daily average concentrations of PM_10_ and PM_2.5_ increased with dust storm intensity, but high levels of SO_2_, NO_2_, O_3,_ and CO concentrations may not be caused by dust storm weather.Dust weather has significant impacts on public health, no matter the age group. The influence of dust weather on different age groups shows that people in the age group above 60 years and below 15 years were more sensitive to dusty weather than people in the age groups between 15 and 60.The results of the PCA further confirmed the relationships between dust storm intensity, air pollutants, meteorological factors, and health symptoms. Higher concentrations of PM_10_ and PM_2.5_s, which were caused by dust storms, were significantly correlated with upper respiratory system diseases and ENT diseases such as “dry throat with bitter taste”, “itchy throat”, “hoarseness”, “dry eyes”, and “stuffy nose”. Some of the symptoms, such as “tears”, “runny nose”, “cough”, and “expectoration”, may not be caused by dust weather, but were likely related to the low temperature and higher concentrations of toxic air pollutants (SO_2_, NO_2_, CO, and O_3_). Further research needs to be conducted, combined with laboratorial analysis.

## Figures and Tables

**Figure 1 ijerph-19-04022-f001:**
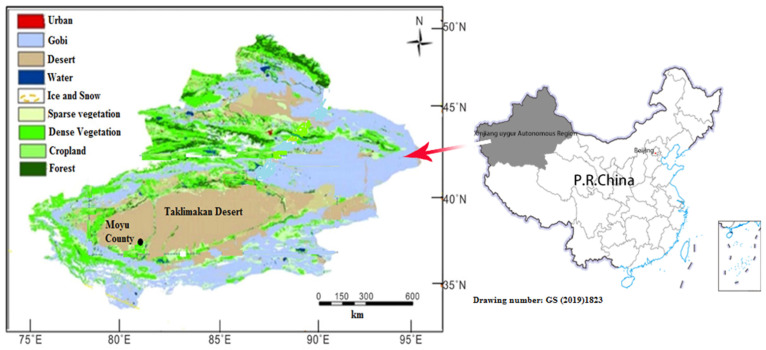
Location and surrounding environment of the study area.

**Figure 2 ijerph-19-04022-f002:**
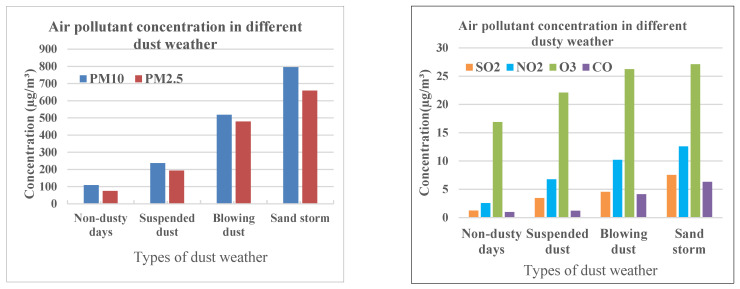
Air pollutant concentrations in different dust weather conditions.

**Figure 3 ijerph-19-04022-f003:**
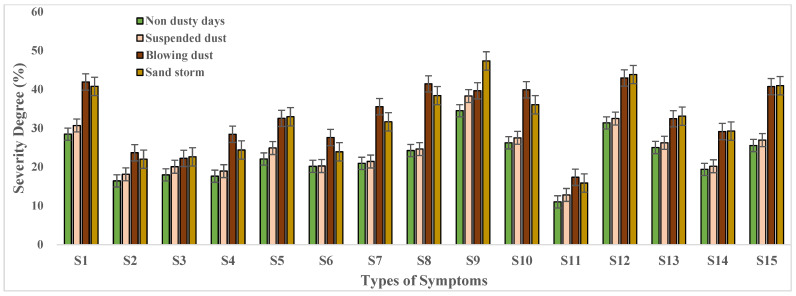
The severity of different health symptoms in different dusty weather conditions. Note: S_1_: dry throat with bitter taste, S_2_: tears, S_3_: runny nose, S_4_: sneeze, S_5_: dry eyes, S_6_: shortness of breath, S_7_: chest tightness, S_8_: cough, S_9_: depression, S_10_: expectoration, S_11_: stuffy nose, S_12_: dry and itchy throat, S_13_: hoarseness, S_14_: cleft lip, and S_15_: mouth ulcer.

**Figure 4 ijerph-19-04022-f004:**
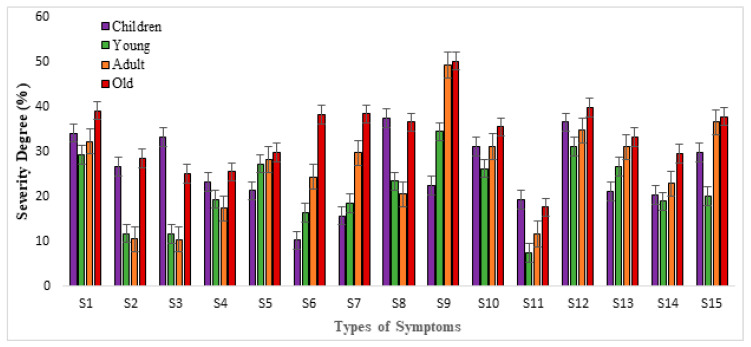
The severity of different health symptoms for different age groups. Note: S_1_: dry throat with bitter taste, S_2_: tears, S_3_: runny nose, S_4_: sneeze, S_5_: dry eyes, S_6_: shortness of breath, S_7_: chest tightness, S_8_: cough, S_9_: depression, S_10_: expectoration, S_11_: stuffy nose, S_12_: dry and itchy throat, S_13_: hoarseness, S_14_: cleft lip, and S_15_: mouth ulcer.

**Table 1 ijerph-19-04022-t001:** Loadings of variables on the selected principal components (normalized variables with Varimax).

Variables	Principal Component (PC)			
	1	2	3	4
Dust storm intensity	**0.644**	**−0.656**	−0.029	0.106
Temperature	**0.544**	0.056	−0.029	**0.529**
Wind speed	**0.752**	−0.235	**0.546**	0.023
Air humidity	0.049	**0.603**	−0.041	−0.017
Air pressure	-0.03	**0.416**	0.02	0.007
PM_10_	**0.419**	−0.041	**0.489**	0.114
PM_2.5_	**0.402**	−0.043	**0.426**	0.092
SO_2_	−0.008	**0.508**	0.209	−0.035
NO_2_	−0.003	**0.516**	0.278	−0.04
CO	0.013	**0.412**	−0.29	−0.102
O_3_	−0.013	0.009	−0.023	**0.523**
Dry throat with bitter taste	**0.471**	0.109	0.007	−0.019
Tears	0.013	**0.616**	0.081	−0.118
Runny nose	−0.034	**0.526**	−0.001	−0.071
Sneeze	0.076	0.073	0.078	**0.478**
Dry eyes	**0.439**	0.119	−0.08	**0.497**
Shortness of breath	0.099	−0.055	**0.405**	−0.037
Chest tightness	0.111	−0.044	**0.545**	−0.041
Cough	0.118	**0.684**	0.02	−0.017
Depression	**0.578**	0.078	−0.157	0.131
Expectoration	0.114	**0.544**	0.027	−0.058
Stuffy nose	0.014	0.093	**0.446**	0.102
Itchy throat	**0.514**	−0.089	−0.021	−0.002
Hoarseness	**0.491**	−0.077	−0.117	0.121
Cleft lip	0.109	−0.027	**0.517**	−0.017
Mouth ulcer	0.105	−0.015	**0.649**	−0.194
Percentage of variance explained	37.6%	14.5%	10.4%	8.8%

Bold shows whose values are above 0.400.

## Data Availability

The data that support the findings of this study are available from the corresponding author, upon reasonable request.

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
