# Peer review of "Health Effects of Dust Storms on the South Edge of the Taklimakan Desert, China: A Survey-Based Approach"

_ijerph, 2022, doi:10.3390/ijerph19074022_

Round 1
Reviewer 1 Report
The authors apply the questionnaire survey method and the compare mean tool to investigate the health effects of the dust storm on public health in Moyu County at the South edge of Taklimakan desert of China. They gained the input data from 1200 respondents for 15 health symptoms, and they got valuable results. The manuscript is of high scientific and particularly data quality. Therefore, I have no major concerns about the manuscript.
However, the paper could gain greater impact if the following problems could be addressed:
- the research questions and the novelty of the work should be clearly stated in the introduction section.
- The literature review is superficial. The authors may consider doing a thorough literature review or a short literature review, especially on health effects in the Taklimakan desert or other deserts.
- When reporting datasets that involved human subjects, authors should declare that the investigations are being approved by an ethics committee before undertaking the research. In this study, 1200 people were interviewed, how did the authors make sure all the respondents were respected in terms of ethical issues?
- The sample size is 1200 in this study, any comments on if it is a good enough number? Or how did the authors calculate the sample size?
- It seems 1200 is the designed sample size, and fortunately, all 1200 people responded to the survey. Did the authors have a plan B if there are some people rejecting the survey?
- There are some abbreviations in the paper, it is suggested that the authors list them in a table at the end of the paper.
- Line 33: can “virus” be categorized into toxic pollutants?
- The authors should be aware of some typos, for example:
1) Line 111: teacher -> teachers;
2) Line 172: Fig. ->Figure;
3) Line 255: 3. Conclusion -> 4.Conclusion.
Reviewer 2 Report
The information shown in the article is very interesting and can be very useful for topics related to the respiratory health of the population of Moyu County. The analysis by PCA allows to discriminate between the different sources of pollutants and the different types of health problems.I think it would be interesting to describe in a simple way what experimental methods are used in the Environmental Monitoring Center of Xinjiang Uyghur Autonomous Region (XUAR) regarding PM2.5 and PM10 (if they are gravimetric, optical methods or by another method). And the same for the other pollutants (gases). Lastly and most importantly, since it exists a clear correlation between dust storms and health problems, it would be interesting to be able to show some kind of prediction of these events to warn the population and warn them that on the critical days they should stay more in their homes or use protective measures such as masks outdoors.
Sincerely yours
Author Response
please see the attachement

Reviewer 3 Report
This article studies some dust storm-induced health issues of different severity amongst different age groups in the Taklimakan Desert. This study is important in the disciplinary regime of public health. Methodology of this article is well designed and the 15 diseases covered in the study are comprehensive. Below are my suggestions:
The respondent sampling approach and demographic selection strategy will be more clear if the authors can expand them a bit.
I think this article lacks a Discussion section; or if editors allow, section 3 can be named Results and Discussion.
Line 135, what does “standardized” here stand for? How these data are pre-processed?
Line 137, how are blowing dust weather and suspended dust weather defined and differentiated? I have this question because based on Lines 156-157, these two categories seem not remarkably different in terms of PM values.
I suggest adding statistical significance bars in Figures 2 and 3 to indicate significant differences. Something like “significant difference” in Line 172 should be supported by statistical analysis.
In Lines 175-176, I don’t see this general trend in Figure 2. But in Figure 2, it is interesting to see that “non dusty days” is similar to “suspended dust weather”, while “blowing dust weather” is similar to “sand storm”. Can the authors interpret this finding a bit?
I assume the severity degree in Figure 2 is identical to severity in Figure 3. But they should be consistent.
The age group (25-60) is worth more discussion since they are also sensitive to dust weather.
If the point to have Table 1 is to discuss correlations, then why the authors do not perform correlate analysis? If use PCA analysis, it is better to show a PCA biplot.
To discuss results, the authors should cite others’ articles to support their discussion points.
I suggest authors revise their language again carefully; the current version reads a bit stilted.
There are also some grammatical errors. For example, in Line 1, “world’s second large shifting sand desert” should be “the world’s second largest shifting sand desert”; in Line 108, “in end of February” should be “at the end of February”, etc.
Author Response
see the attachement

Round 2
Reviewer 3 Report
Reviewer 3
The authors have addressed most of my previous comments well. But still a few more points below to revise:
Point 5: I still think some mean comparison statistical analyses like t-test and ANOVA should be performed to support the visual difference.
Point 7: I did not say the two figures were the same. I meant the titles of y-axes of Figures 3 and 4 should be kept the same.
Point 10: If the authors changed the “Results” section to “Results and Discussion”, then citations and comparisons with others’ work should be discussed through this section, instead of just the final two paragraphs.
Point 11: The language of the current version is still a bit stilted. Can be better. And something like “Discussions” in the section title should be “Discussion” (singular). A lot more formatting problems in the whole text. Please reorganize.
Author Response
Please see attachement
